# Manifold Structured Prediction

**Alessandro Rudi** [•,1]    **Carlo Ciliberto** [•,*,2]    **Gian Maria Marconi** [3]    **Lorenzo Rosasco** [3,4]

`alessandro.rudi@inria.fr  c.ciliberto@imperial.ac.uk  gian.maria.marconi@iit.it  lrosasco@mit.edu`

[1]INRIA - Département d'informatique, École Normale Supérieure - PSL Research University, Paris, France.
[2]Department of Electrical and Electronic Engineering, Imperial College, London, UK.
[3]Università degli studi di Genova & Istituto Italiano di Tecnologia, Genova, Italy.
[4]Massachusetts Institute of Technology, Cambridge, USA.
[•] Equal Contribution

## Abstract

Structured prediction provides a general framework to deal with supervised problems where the outputs have semantically rich structure. While classical approaches consider finite, albeit potentially huge, output spaces, in this paper we discuss how structured prediction can be extended to a continuous scenario. Specifically, we study a structured prediction approach to manifold valued regression. We characterize a class of problems for which the considered approach is statistically consistent and study how geometric optimization can be used to compute the corresponding estimator. Promising experimental results on both simulated and real data complete our study.

## 1   Introduction

Regression and classification are probably the most classical machine learning problems and correspond to estimating a function with scalar and binary values, respectively. In practice, it is often interesting to estimate functions with more structured outputs. When the output space can be assumed to be a vector space, many ideas from regression can be extended, think for example to multivariate [20] or functional regression [32]. However, a lack of a natural vector structure is a feature of many practically interesting problems, such as ranking [18], quantile estimation [26] or graph prediction [38]. In this latter case, the outputs are typically provided only with some distance or similarity function that can be used to design appropriate loss function. Knowledge of the loss is sufficient to analyze an abstract empirical risk minimization approach within the framework of statistical learning theory, but deriving approaches that are at the same time statistically sound and computationally feasible is a key challenge. While ad-hoc solutions are available for many specific problems [15, 37, 24, 7], structured prediction [5] provides a unifying framework where a variety of problems can be tackled as special cases.

Classically, structured prediction considers problems with finite, albeit potentially huge, output spaces. In this paper, we study how these ideas can be applied to non-discrete output spaces. In particular, we consider the case where the output space is a Riemannian manifold, that is the problem of manifold structured prediction (also called manifold valued regression [46]). While also in this case ad-hoc methods are available [47], in this paper we adopt and study a structured prediction approach starting from a framework proposed in [13]. Within this framework, it is possible to derive a statistically sound, and yet computationally feasible, structured prediction approach as long as the loss function satisfies suitable structural assumptions [14, 17, 29, 25, 12, 36]. Moreover we can guarantee that the computed prediction is always an element of the manifold.

Our main technical contribution is a characterization of loss functions for manifold structured prediction satisfying such a structural assumption. In particular, we consider the case where the

---

[*]Work performed while C.C. was at the University College London.

Riemannian metric is chosen as a loss function. As a byproduct of these results, we derive a manifold structured learning algorithm that is universally consistent and corresponding finite sample bounds. From a computational point of view, the proposed algorithm requires solving a linear system (at training time) and a minimization problem over the output manifold (at test time). To tackle this latter problem, we investigate the application of geometric optimization methods, and in particular Riemannian gradient descent [1]. We consider both numerical simulations and benchmark datasets reporting promising performances. The rest of the paper is organized as follows. In Section 2, we define the problem and explain the proposed algorithm. In Section 3 we state and prove the theoretical results of this work. In Section 4 we explain how to compute the proposed algorithm and we show the performance of our method on synthetic and real data.

## 2 Structured Prediction for Manifold Valued Regression

The goal of supervised learning is to find a functional relation between an input space $\mathcal{X}$ and an output space $\mathcal{Y}$ given a finite set of observations. Traditionally, the output space is either a linear space (e.g. $\mathcal{Y} = \mathbb{R}^M$) or a discrete set (e.g. $\mathcal{Y} = \{0, 1\}$ in binary classification). In this paper, we consider the problem of manifold structured prediction [47], in which output data lies on a manifold $\mathcal{M} \subset \mathbb{R}^d$. In this context, statistical learning corresponds to solving

$$\underset{f \in \mathcal{X} \to \mathcal{Y}}{\operatorname{argmin}} \mathcal{E}(f) \qquad \text{with} \qquad \mathcal{E}(f) = \int_{\mathcal{X} \times \mathcal{Y}} \triangle(f(x), y)\, \rho(x, y) \tag{1}$$

where $\mathcal{Y}$ is a subset of the manifold $\mathcal{M}$ and $\rho$ is an unknown distribution on $\mathcal{X} \times \mathcal{Y}$. Here, $\triangle : \mathcal{Y} \times \mathcal{Y} \to \mathbb{R}$ is a loss function that measures prediction errors for points estimated on the manifold. The minimization is meant over the set of all measurable functions from $\mathcal{X}$ to $\mathcal{Y}$. The distribution is fixed but unknown and a learning algorithm seeks an estimator $\widehat{f} : \mathcal{X} \to \mathcal{Y}$ that approximately solves Eq. (1), given a set of training points $(x_i, y_i)_{i=1}^n$ sampled independently from $\rho$.

A concrete example of loss function that we will consider in this paper is $\triangle = d^2$ the squared geodesic distance $d : \mathcal{Y} \times \mathcal{Y} \to \mathbb{R}$ [27]. The geodesic distance is the natural metric on a Riemannian manifold (it corresponds to the Euclidean distance when $\mathcal{M} = \mathbb{R}^d$) and is a natural loss function in the context of manifold regression [46, 47, 19, 23, 21].

### 2.1 Manifold Valued Regression via Structured Prediction

In this paper we consider a structured prediction approach to manifold valued regression following ideas in [13]. Given a training set $(x_i, y_i)_{i=1}^n$, an estimator for problem Eq. (1) is defined by

$$\widehat{f}(x) = \underset{y \in \mathcal{Y}}{\operatorname{argmin}} \sum_{i=1}^n \alpha_i(x) \triangle (y, y_i) \tag{2}$$

for any $x \in \mathcal{X}$. The coefficients $\alpha(x) = (\alpha_1(x), \ldots, \alpha_n(x))^\top \in \mathbb{R}^n$ are obtained solving a linear system for a problem akin to kernel ridge regression (see Section 2.2): given a positive definite kernel $k : \mathcal{X} \times \mathcal{X} \to \mathbb{R}$ [4] over $\mathcal{X}$, we have

$$\alpha(x) = (\alpha_1(x), \ldots, \alpha_n(x))^\top = (K + n\lambda I)^{-1} K_x \tag{3}$$

where $K \in \mathbb{R}^{n \times n}$ is the empirical kernel matrix with $K_{i,j} = k(x_i, x_j)$, and $K_x \in \mathbb{R}^n$ the vector whose $i$-th entry corresponds to $(K_x)_i = k(x, x_i)$. Here, $\lambda \in \mathbb{R}_+$ is a regularization parameter and $I \in \mathbb{R}^{n \times n}$ denotes identity matrix.

Computing the estimator in Eq. (2) can be divided into two steps. During a *training step* the score function $\alpha : \mathcal{X} \to \mathbb{R}^n$ is learned, while during the *prediction step*, the output $\widehat{f}(x) \in \mathcal{Y}$ is estimated on a new test point $x \in \mathcal{X}$. This last step requires minimizing the linear combination of distances $\triangle(y, y_i)$ between a candidate $y \in \mathcal{Y}$ and the training outputs $(y_i)_{i=1}^n$, weighted by the corresponding scores $\alpha_i(x)$. Next, we recall the derivation of the above estimator following [13].

### 2.2 Derivation of the Proposed Estimator

The derivation of $\widehat{f}$ in Eq. (2) is based on the following key structural assumption on the loss.

**Definition 1** (Structure Encoding Loss Function (SELF)). *Let $\mathcal{Y}$ be a compact set. A function $\triangle : \mathcal{Y} \times \mathcal{Y} \to \mathbb{R}$ is a Structure Encoding Loss Function if there exist a separable Hilbert space $\mathcal{H}$, a continuous feature map $\psi : \mathcal{Y} \to \mathcal{H}$ and a continuous linear operator $V : \mathcal{H} \to \mathcal{H}$ such that for all $y, y' \in \mathcal{Y}$*

$$\triangle(y, y') = \langle \psi(y), V\psi(y') \rangle_{\mathcal{H}}. \tag{4}$$

Intuitively, the SELF definition requires a loss function to be "bi-linearizable" over the space $\mathcal{H}$. This is similar, but more general, than requiring the loss to be a kernel since it allows also to consider distances (which are not positive definite) or even non-symmetric loss functions. As observed in [13], a wide range of loss functions often used in machine learning are SELF. In Section 3 we study how the above assumption applies to manifold structured loss functions, including the squared geodesic distance.

We first recall how the estimator Eq. (2) can be obtained assuming $\triangle$ to be SELF. We begin by rewriting the expected risk in Eq. (1) as

$$\mathcal{E}(f) = \int_{\mathcal{X}} \left\langle \psi(f(x)), V \int_{\mathcal{Y}} \psi(y) \, d\rho(y|x) \right\rangle_{\mathcal{H}} d\rho(x) \tag{5}$$

where we have conditioned $\rho(y, x) = \rho(y|x)\rho_{\mathcal{X}}(x)$ and used the linearity of the integral and the inner product. Therefore, any function $f^* : \mathcal{X} \to \mathcal{Y}$ minimizing the above functional must satisfy the following condition

$$f^*(x) = \underset{y \in \mathcal{Y}}{\operatorname{argmin}} \langle \psi(y), Vg^*(x) \rangle_{\mathcal{H}} \qquad \text{where} \qquad g^*(x) = \int_{\mathcal{Y}} \psi(y) \, d\rho(y|x) \tag{6}$$

where we have introduced the function $g^* : \mathcal{X} \to \mathcal{H}$ that maps each point $x \in \mathcal{X}$ to the conditional expectation of $\psi(y)$ given $x$. However we cannot compute explicitly $g^*$, but noting that it minimizes the expected least squares error

$$\int \|\psi(y) - g(x)\|_{\mathcal{H}}^2 d\rho(x, y) \tag{7}$$

suggests that a least squares estimator can be considered. We first illustrate this idea for $\mathcal{X} = \mathbb{R}^d$ and $\mathcal{H} = \mathbb{R}^k$. In this case we can consider a ridge regression estimator

$$\widehat{g}(x) = \widehat{W}^{\top} x \qquad \text{with} \qquad \widehat{W} = \underset{W \in \mathbb{R}^{d \times k}}{\operatorname{argmin}} \frac{1}{n} \|XW - \psi(Y)\|_F^2 + \lambda \|W\|_F^2 \tag{8}$$

where $X = (x_1, \ldots, x_n)^{\top} \in \mathbb{R}^{n \times d}$ and $\psi(Y) = (\psi(y_1), \ldots, \psi(y_n))^{\top} \in \mathbb{R}^{n \times k}$ are the matrices whose $i$-th row correspond respectively to the training sample $x_i \in \mathcal{X}$ and the (mapped) training output $\psi(y_i) \in \mathcal{H}$. We have denoted $\| \cdot \|_F^2$ the squared Frobenius norm of a matrix, namely the sum of all its squared entries. The ridge regression solution can be obtained in closed form as $\widehat{W} = (X^{\top} X + n\lambda I)^{-1} X^{\top} \psi(Y)$. For any $x \in \mathcal{X}$ we have

$$\widehat{g}(x) = \psi(Y)^{\top} X (X^{\top} X + n\lambda I)^{-1} x = \psi(Y)^{\top} \alpha(x) = \sum_{i=1}^{n} \alpha_i(x) \psi(y_i) \tag{9}$$

where we have introduced the coefficients $\alpha(x) = X(X^{\top} X + n\lambda I)^{-1} x \in \mathbb{R}^n$. By substituting $\widehat{g}$ to $g^*$ in Eq. (6) we have

$$\widehat{f}(x) = \underset{y \in \mathcal{M}}{\operatorname{argmin}} \left\langle \psi(y), V \left( \sum_{i=1}^{n} \alpha_i(x) \psi(y_i) \right) \right\rangle = \underset{y \in \mathcal{M}}{\operatorname{argmin}} \sum_{i=1}^{n} \alpha_i(x) \triangle (y, y_i) \tag{10}$$

where we have used the linearity of the sum and the inner product to move the coefficients $\alpha_i$ outside of the inner product. Since the loss is SELF, we then obtain $\langle \psi(y), V\psi(y_i) \rangle = \triangle(y, y_i)$ for any $y_i$ in the training set. This recovers the estimator $\widehat{f}$ introduced in Eq. (2), as desired.

We end noting how the above idea can be extended. First, we can consider $\mathcal{X}$ to be a set and $k : \mathcal{X} \times \mathcal{X} \to \mathbb{R}$ a positive definite kernel. Then $\widehat{g}$ can be computed by kernel ridge regression (see e.g. [43]) to obtain the scores $\alpha(x) = (K + n\lambda I)^{-1} K_x$, see Eq. (3). Second, the above discussion applies if $\mathcal{H}$ is infinite dimensional. Indeed, thanks to the SELF assumption, $\widehat{f}$ does not depend on explicit knowledge of the space $\mathcal{H}$ but only on the loss function.

We next discuss the main results of the paper, showing that a large class of loss functions for manifold structured prediction are SELF. This will allow us to prove consistency and learning rates for the manifold structured estimator considered in this work.

# 3   Characterization of SELF Function on Manifolds

In this section we provide sufficient conditions for a wide class of functions on manifolds to satisfy the definition of SELF. A key example will be the case of the squared geodesic distance. To this end we will make the following assumptions on the manifold $\mathcal{M}$ and the output space $\mathcal{Y} \subseteq \mathcal{M}$ where the learning problem takes place.

**Assumption 1.** *$\mathcal{M}$ is a complete $d$-dimensional smooth connected Riemannian manifold, without boundary, with Ricci curvature bounded below and positive injectivity radius.*

The assumption above imposes basic regularity conditions on the output manifold. In particular we require the manifold to be locally diffeomorphic to $\mathbb{R}^d$ and that the tangent space of $\mathcal{M}$ at any $p \in \mathcal{M}$ varies smoothly with respect to $p$. This assumption avoids pathological manifolds and is satisfied for instance by any smooth compact manifold (e.g. the sphere, torus, etc.) [27]. Other notable examples are the statistical manifold (without boundary) [3] any open bounded sub-manifold of the cone of positive definite matrices, which is often studied in geometric optimization settings [1]. This assumption will be instrumental to guarantee the existence of a space of functions $\mathcal{H}$ on $\mathcal{M}$ rich enough to contain the squared geodesic distance.

**Assumption 2.** *$\mathcal{Y}$ is a compact geodesically convex subset of the manifold $\mathcal{M}$.*

A subset $\mathcal{Y}$ of a manifold is geodesically convex if for any two points in $\mathcal{Y}$ there exists one and only one minimizing geodesic curve connecting them. The effect of Asm. 2 is twofold. On one hand it guarantees a generalized notion of convexity for the space $\mathcal{Y}$ on which we will solve the optimization problem in Eq. (2). On the other hand it avoids the geodesic distance to have singularities on $\mathcal{Y}$ (which is key to our main result below). For a detailed introduction to most definitions and results reviewed in this section we refer the interested reader to standard references for differential and Riemannian geometry (see e.g. [27]). We are ready to prove the main result of this work.

**Theorem 1** (Smooth Functions are SELF). *Let $\mathcal{M}$ satisfy Asm. 1 and $\mathcal{Y} \subseteq \mathcal{M}$ satisfy Asm. 2. Then, any smooth function $h : \mathcal{Y} \times \mathcal{Y} \to \mathbb{R}$ is SELF on $\mathcal{Y}$.*

*Sketch of the proof (Thm. 1).* The complete proof of Thm. 1 is reported in **??**. The proof hinges around the following key steps:

**Step 1 If there exists an RKHS $\mathcal{H}$ on $\mathcal{M}$, then any $h \in \mathcal{H} \otimes \mathcal{H}$ is SELF.** Let $\mathcal{H}$ be a reproducing kernel Hilbert space (RKHS) [4] of functions on $\mathcal{M}$ with associated bounded kernel $k : \mathcal{M} \times \mathcal{M} \to \mathbb{R}$. Let $\mathcal{H} \otimes \mathcal{H}$ denote the RKHS of functions $h : \mathcal{M} \times \mathcal{M} \to \mathbb{R}$ with associated kernel $\bar{k}$ such that $\bar{k}((y,z),(y',z')) = k(y,y')k(z,z')$ for any $y, y', z, z' \in \mathcal{M}$. Let, $h : \mathcal{M} \times \mathcal{M} \to \mathbb{R}$ be such that $h \in \mathcal{H} \otimes \mathcal{H}$. Recall that $\mathcal{H} \otimes \mathcal{H}$ is isometric to the space of Hilbert-Schmidt operators from $\mathcal{H}$ to itself. Let $V_h : \mathcal{H} \to \mathcal{H}$ be the operator corresponding to $h$ via such isometry. We show that the SELF definition is satisfied with $V = V_h$ and $\psi(y) = k(y, \cdot) \in \mathcal{H}$ for any $y \in \mathcal{M}$. In particular, we have $\|V\| \leq \|V\|_{\mathrm{HS}} = \|h\|_{\mathcal{H} \otimes \mathcal{H}}$, with $\|V\|_{\mathrm{HS}}$ denoting the Hilbert-Schmidt norm of $V$.

**Step 2: Under Asm. 2, $C_0^\infty(\mathcal{M}) \otimes C_0^\infty(\mathcal{M})$ "contains" $C^\infty(\mathcal{Y} \times \mathcal{Y})$.** If $\mathcal{Y}$ is compact and geodesically convex, then it is diffeomorphic to a compact set of $\mathbb{R}^d$. By using this fact, we prove that any function in $C^\infty(\mathcal{Y} \times \mathcal{Y})$, the space of smooth functions on $\mathcal{Y} \times \mathcal{Y}$, admits an extension in $C_0^\infty(\mathcal{M} \times \mathcal{M})$ the space of smooth functions on $\mathcal{M} \times \mathcal{M}$ vanishing at infinity (this is well defined since $\mathcal{M}$ is diffeomorphic to $\mathbb{R}^d$, see **??** in the supplementary material), and that $C_0^\infty(\mathcal{M} \times \mathcal{M}) = C_0^\infty(\mathcal{M}) \otimes C_0^\infty(\mathcal{M})$, with $\otimes$ the canonical topological tensor product [50].

**Step 3: Under Asm. 1, there exists an RKHS on $\mathcal{M}$ containing $C_0^\infty(\mathcal{M})$.** Under Asm. 1, the Sobolev space $\mathcal{H} = H_s^2(\mathcal{M})$ of square integrable functions with smoothness $s$ is an RKHS for any $s > d/2$ (see [22] for a definition of Sobolev spaces on Riemannian manifolds).

The proof proceeds as follows: from Step 1, we see that to guarantee $h$ to be SELF it is sufficient to prove the existence of an RKHS $\mathcal{H}$ such that $h \in \mathcal{H} \otimes \mathcal{H}$. The rest of the proof is therefore devoted to showing that for smooth functions this is satisfied for $\mathcal{H} = H_s^2(\mathcal{M})$. Since $h$ is smooth, by Step 2 we have that under Asm. 2, there exists a $\bar{h} \in C_0^\infty(\mathcal{M}) \otimes C_0^\infty(\mathcal{M})$ whose restriction $\bar{h}|_{\mathcal{Y} \times \mathcal{Y}}$ to $\mathcal{Y} \times \mathcal{Y}$ corresponds to $h$. Now, denote by $H_s^2(\mathcal{M})$ the Sobolev space of squared integrable functions on $\mathcal{M}$ with smoothness index $s > 0$. By construction, (see [22]) for any $s > 0$, we have $C_0^\infty(\mathcal{M})|_{\mathcal{Y}} \subseteq H_s^2(\mathcal{M})|_{\mathcal{Y}}$, namely for any function. In particular, $\bar{h} \in C_0^\infty(\mathcal{M}) \otimes C_0^\infty(\mathcal{M}) \subseteq H_s^2(\mathcal{M}) \otimes H_s^2(\mathcal{M})$. Finally, Step 3 guarantees that under Asm. 1, $\mathcal{H} = H_s^2(\mathcal{M})$ with $s > d/2$ is an RKHS, showing that $h \in \mathcal{H} \otimes \mathcal{H}$ as desired. $\square$

Interestingly, Thm. 1 shows that the SELF estimator proposed in Eq. (2) can tackle *any* manifold valued learning problem in the form of Eq. (1) with smooth loss function. In the following we study the specific case of the squared geodesic distance.

**Theorem 2** ($d^2$ is SELF). *Let $\mathcal{M}$ satisfy Asm. 1 and $\mathcal{Y} \subseteq \mathcal{M}$ satisfy Asm. 2. Then, the squared geodesic distance $\triangle = d^2 : \mathcal{M} \times \mathcal{M} \to \mathbb{R}$ is smooth on $\mathcal{Y}$. Therefore $\triangle$ is SELF on $\mathcal{Y}$.*

The proof of the result above is reported in the supplementary material. The main technical aspect is to show that regularity provided by Asm. 2 guarantees the squared geodesic distance to be smooth. The fact that $\triangle$ is SELF is then an immediate corollary of Thm. 1.

### 3.1 Statistical Properties of Manifold Structured Prediction

In this section, we characterize the generalization properties of the manifold structured estimator Eq. (2) in light of Thm. 1 and Thm. 2.

**Theorem 3** (Universal Consistency). *Let $\mathcal{M}$ satisfy Asm. 1 and $\mathcal{Y} \subseteq \mathcal{M}$ satisfy Asm. 2. Let $\mathcal{X}$ be a compact set and $k : \mathcal{X} \times \mathcal{X} \to \mathbb{R}$ be a bounded continuous universal kernel[2] For any $n \in \mathbb{N}$ and any distribution $\rho$ on $\mathcal{X} \times \mathcal{Y}$ let $\widehat{f}_n : \mathcal{X} \to \mathcal{Y}$ be the manifold structured estimator in Eq. (2) for a learning problem with smooth loss function $\triangle$, with $(x_i, y_i)_{i=1}^n$ training points independently sampled from $\rho$ and $\lambda_n = n^{-1/4}$. Then*

$$\lim_{n \to \infty} \mathcal{E}(\widehat{f}_n) = \mathcal{E}(f^*) \quad \text{with probability } 1. \tag{11}$$

The result above follows from Thm. 4 in [13] combined with our result in Thm. 1. It guarantees that the algorithm considered in this work finds a consistent estimator for the manifold structured problem, when the loss function is smooth (thus also in the case of the squared geodesic distance). As it is standard in statistical learning theory, we can impose regularity conditions on the learning problem, in order to derive also generalization bounds for $\widehat{f}$. In particular, if we denote by $\mathcal{F}$ the RKHS associated to the kernel $k$, we will require $g^*$ to belong to the same space $\mathcal{H} \otimes \mathcal{F}$ where the estimator $\widehat{g}$ introduced in Eq. (9) is learned. In the simplified case discussed in Section 2.2, with linear kernel on $\mathcal{X} = \mathbb{R}^d$ and $\mathcal{H} = \mathbb{R}^k$ finite dimensional, we have $\mathcal{F} = \mathbb{R}^d$ and this assumption corresponds to require the existence of a matrix $W_*^\top \in \mathbb{R}^{k \times d} = \mathcal{H} \otimes \mathcal{F}$, such that $g^*(x) = W_*^\top x$ for any $x \in \mathcal{X}$. In the general case, the space $\mathcal{H} \otimes \mathcal{F}$ extends to the notion of *reproducing kernel Hilbert space for vector-valued functions* (see e.g. [30, 2]) but the same intuition applies [28, 10, 49].

**Theorem 4** (Generalization Bounds). *Let $\mathcal{M}$ satisfy Asm. 1 and $\mathcal{Y} \subseteq \mathcal{M}$ satisfy Asm. 2. Let $\mathcal{H} = H_s^2(\mathcal{M})$ with $s > d/2$ and $k : \mathcal{X} \times \mathcal{X} \to \mathbb{R}$ be a bounded continuous reproducing kernel with associated RKHS $\mathcal{F}$. For any $n \in \mathbb{N}$, let $\widehat{f}_n$ denote the manifold structured estimator in Eq. (2) for a learning problem with smooth loss $\triangle : \mathcal{Y} \times \mathcal{Y} \to \mathbb{R}$ and $\lambda_n = n^{-1/2}$. If the conditional mean $g^*$ belongs to $\mathcal{H} \otimes \mathcal{F}$, then*

$$\mathcal{E}(\widehat{f}_n) - \mathcal{E}(f^*) \leq \mathsf{c}_\triangle \mathsf{q} \, \tau^2 \, n^{-\frac{1}{4}} \tag{12}$$

*holds with probability not less than $1 - 8e^{-\tau}$ for any $\tau > 0$, with $\mathsf{c}_\triangle = \| \triangle \|_{\mathcal{H} \otimes \mathcal{H}}$ and $\mathsf{q}$ a constant not depending on $n, \tau$ or the loss $\triangle$.*

The generalization bound of Thm. 4 is obtained by adapting Thm. 5 of [13] to our results in Thm. 1 as detailed in the supplementary material. To our knowledge these are the first results characterizing in such generality the generalization properties of an estimator for manifold structured learning with generic smooth loss function. We conclude with a remark on a key quantity in the bound of Thm. 4.

**Remark 1** (The constant $\mathsf{c}_\triangle$). *We comment on the role played in the learning rate by $\mathsf{c}_\triangle$, the norm of the loss function $\triangle$ seen an element of the Hilbert space $\mathcal{H} \otimes \mathcal{H}$. Indeed, from the discussion of Thm. 1 we have seen that any smooth function on $\mathcal{Y}$ is SELF and belongs to the set $\mathcal{H} \otimes \mathcal{H}$ with $\mathcal{H} = H_s^2(\mathcal{M})$, the Sobolev space of squared integrable functions for $s > d/2$. Following this interpretation, we see that the bound in Thm. 4 can improve significantly (in terms of the constants) depending on the regularity of the loss function: smoother loss functions will result in "simpler" learning problems and vice-versa. In particular, when $\triangle$ corresponds to the squared geodesic distance, the more "regular" is the manifold $\mathcal{M}$, the learning problem will be. A refined quantitative characterization of $\mathsf{c}_\triangle$ in terms of the Ricci curvature and the injective radius of the manifold is left to future work.*

Table 1: Structured loss, gradient of the structured loss and retraction for $P_{++}^m$ and $S_{d-1}$. $Z_i \in P_{++}^m$ and $z_i \in S_{d-1}$ are the training set points. $I \in \mathbb{R}^{d \times d}$ is the identity matrix.

| | Positive definite matrix manifold $(P_{++}^m)$ | Sphere $(S_{d-1})$ |
|---|---|---|
| $F(y)$ | $\sum_{i=1}^{n} \alpha_i \| \log(Y^{-\frac{1}{2}} Z_i Y^{-\frac{1}{2}}) \|_F^2$ | $\sum_{i=1}^{n} \alpha_i \arccos\left( \langle z_i, y \rangle \right)^2$ |
| $\nabla_{\mathcal{M}} F(y)$ | $2 \sum_{i=1}^{n} \alpha_i Y^{\frac{1}{2}} \log(Y^{\frac{1}{2}} Z_i^{-1} Y^{\frac{1}{2}}) Y^{\frac{1}{2}}$ | $4 \sum_{i=1}^{n} \alpha_i (yy^T - I) \frac{\arccos(\langle z_i, y \rangle)}{\sqrt{1 - \langle z_i, y \rangle}} z_i$ |
| $R_y(v)$ | $Y^{\frac{1}{2}} \exp(Y^{-\frac{1}{2}} v Y^{-\frac{1}{2}}) Y^{\frac{1}{2}}$ | $\frac{v}{\|v\|}$ |

## 4 Manifold Structured Prediction Algorithm and Experiments

In this section we recall geometric optimization algorithms that can be adopted to perform the estimation of $\widehat{f}$ on a novel test point $x$. We then evaluate the performance of the proposed method in practice, reporting numerical results on simulated and real data.

The algorithm presented in this paper consists in two steps. In the *training* phase, the matrix $C = (K + \lambda n I)^{-1}$ is computed, requiring a computational cost of essentially $O(n^3)$ in time and $O(n^2)$ in space (see Eq. (3)). In the *inference* phase, given a test point $x$, the coefficients in Eq. (3) are computed, $\alpha(x) := Cv(x)$, requiring essentially $O(n)$ in time, and then the optimization problem in Eq. (2) is solved (see next subsection for more details). Note that it is possible to reduce the computational complexity of the training and evaluation of the coefficients, while retaining the statistical guarantees of the proposed method. Indeed the computation of the coefficients consists essentially in solving a kernel ridge regression problem [10] as analyzed in [13], for which methods based on random projection, as *Nyström* [44] or *random features* [39], reduce the complexity up to $O(n\sqrt{n})$ in time and $O(n)$ in space, while guaranteeing the same statistical properties [40, 42, 9, 11, 41].

### 4.1 Optimization on Manifolds

We begin discussing the computational aspects related to evaluating the manifold structured estimator. In particular, we discuss how to address the optimization problem in Eq. (2) in specific settings. Given a test point $x \in \mathcal{X}$, this process consists in solving a minimization over $\mathcal{Y}$, namely

$$\min_{y \in \mathcal{Y}} F(y) \tag{13}$$

where $F(y)$ corresponds to the linear combination of $\triangle(y, y_i)$ weighted by the scores $\alpha_i(x)$ computed according to Eq. (3). If $\mathcal{Y}$ is a linear manifold or a subset of $\mathcal{M} = \mathbb{R}^d$, this problem can be solved by means of gradient-based minimization algorithms, such as Gradient Descent (GD):

$$y_{t+1} = y_t - \eta_t \nabla F(y_t) \tag{14}$$

for a step size $\eta_t \in \mathbb{R}$. This algorithm can be extended to *Riemannian gradient descent* (RGD) [52] on manifolds, as

$$y_{t+1} = Exp_{y_t}(\eta_t \nabla_{\mathcal{M}} F(y_t)) \tag{15}$$

Where $\nabla_{\mathcal{M}} F$ is the gradient defined with respect to the Riemannian metric (see [1]) and $Exp_y : T_y \mathcal{M} \to \mathcal{M}$ denotes the exponential map on $y \in \mathcal{Y}$, mapping a vector from the tangent space $T_y \mathcal{M}$ to the associated point on the manifold according to the Riemannian metric [27]. For completeness, the algorithm is recalled in **??**. For this family of gradient-based algorithms it is possible to substitute the exponential map with a retraction $R_y : T_y \mathcal{M} \to \mathcal{M}$, which is a first order approximation to the exponential map. Retractions are often faster to compute and still offer convergence guarantees [1]. In the following experiments we will use both retractions and exponential maps. We mention that the step size $\eta_t$ can be found with a line search over the validation set, for more see [1]. Note that also stochastic optimization algorithms have been generalized to the Riemannian setting such as R-SGD [8]. Interestingly, methods such as R-SGD can be advantageous in our setting. Indeed, solving the inference in Eq. (2) requires solving the minimization of a sum over $n$ elements, it might be favorable from the computational perspective to adopt a stocastic method that at each iteration minimizes the functional with respect to a single (or a mini-batch) of randomly sampled elements of the entire sum.

Table 2: Simulation experiment: average squared loss (First two columns) and $\triangle_{\mathrm{PD}}$ (Last two columns) error of the proposed structured prediction (SP) approach and the KRLS baseline on learning the inverse of a PD matrix for increasing matrix dimension.

| Dim | Squared loss | | $\triangle_{\mathrm{PD}}$ loss | |
|---|---|---|---|---|
| | KRLS | SP | KRLS | SP |
| 5 | 0.72±0.08 | 0.89±0.08 | 111±64 | 0.94±0.06 |
| 10 | 0.81±0.03 | 0.92±0.05 | 44±8.3 | 1.24±0.06 |
| 15 | 0.83±0.03 | 0.91±0.06 | 56±10 | 1.25±0.05 |
| 20 | 0.85±0.02 | 0.91±0.03 | 59±12 | 1.33±0.03 |
| 25 | 0.87±0.01 | 0.91±0.02 | 72±9 | 1.44±0.03 |
| 30 | 0.88±0.01 | 0.91±0.02 | 67±7.2 | 1.55±0.03 |

Table 1 reports gradients and retraction maps for the geodesic distance of two problems of interest considered in this work: positive definite manifold and the sphere. See Sections 4.2 and 4.3 for more details on the related manifolds.

We point out that using optimization algorithms that comply with the geometry of the manifold, such as RGD, guarantees that the computed value is an element of the manifold. This is in contrast with algorithms that compute a solution in a linear space that contains $\mathcal{M}$ and then need to project the computed solution onto $\mathcal{M}$. We next discuss empirical evaluations of the proposed manifold structured estimator on both synthetic and real datasets.

## 4.2 Synthetic Experiments: Learning Positive Definite Matrices

We consider the problem of learning a function $f : \mathbb{R}^d \to \mathcal{Y} = P_{++}^m$, where $P_{++}^m$ denotes the *cone of positive definite (PD)* $m \times m$ matrices. Note that $P_{++}^m$ is a manifold with squared geodesic distance $\triangle_{\mathrm{PD}}$ between any two PD matrices $Z, Y \in P_{++}^m$ defined as

$$\triangle_{\mathrm{PD}}(Z, Y) = \| \log(Y^{-\frac{1}{2}} Z \, Y^{-\frac{1}{2}}) \|_F^2 \qquad (16)$$

where, for any $M \in P_{++}^m$, we have that $M^{\frac{1}{2}}$ and $\log(M)$ correspond to the matrices with same eigenvectors of $M$ but with respectively the square root and logarithm of the eigenvalues of $M$. In Table 1 we show the computation of the structured loss, the gradient of the structured loss and the exponential map of the PD matrix manifold. We refer the reader to [31, 6] for a more detailed introduction on the manifold of positive definite matrices.

For the experiments reported in the following we compared the performance of the manifold structured estimator minimizing the loss $\triangle_{\mathrm{PD}}$ and a Kernel Regularized Least Squares classifier (KRLS) baseline (see Appendix **??**), both trained using the Gaussian kernel $k(x, x') = \exp(-\|x - x'\|^2 / 2\sigma^2)$. The matrices predicted by the KRLS estimator are projected on the PD manifold by setting to a small positive constant ($1e - 12$) the negative eigenvalues. For the manifold structured estimator, the optimization problem at Eq. (2) was performed with the Riemannian Gradient Descent (RGD) algorithm [1]. We refer to [52] regarding the implementation of the RGD in the case of the geodesic distance on the PD cone.

**Learning the Inverse of a Positive Definite Matrix.** We consider the problem of learning the function $f : P_{++}^m \to P_{++}^m$ such that $f(X) = X^{-1}$ for any $X \in P_{++}^m$. Input matrices are generated as $X_i = U \Sigma U^\top \in P_{++}^m$ with $U$ a random orthonormal matrix sampled from the Haar distribution [16] and $S \in P_{++}^m$ a diagonal matrix with entries randomly sampled from the uniform distribution on $[0, 10]$. We generated datasets of increasing dimension $m$ from 5 to 50, each with 1000 points for training, 100 for validation and 100 for testing. The kernel bandwidth $\sigma$ was chosen and the regularization parameter $\lambda$ were selected by cross-validation respectively in the ranges 0.1 to 1000 and $10^{-6}$ to 1 (logarithmically spaced).

Table 2 reports the performance of the manifold structured estimator (SP) and the KRLS baseline with respect to both the $\triangle_{\mathrm{PD}}$ loss and the least squares loss (normalized with respect to the number of dimensions). Note that the KRLS estimator target is to minimize the least squares (Frobenius) loss and is not designed to capture the geometry of the PD cone. We notice that the proposed approach significantly outperforms the KRLS baseline with respect to the $\triangle_{\mathrm{PD}}$ loss. This is expected: $\triangle_{\mathrm{PD}}$

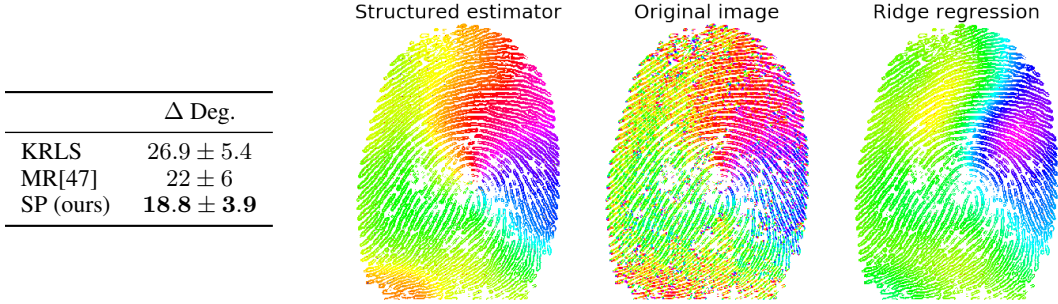

| | $\triangle$ Deg. |
|---|---|
| KRLS | $26.9 \pm 5.4$ |
| MR[47] | $22 \pm 6$ |
| SP (ours) | $\mathbf{18.8 \pm 3.9}$ |

Figure 1: (Left) Fingerprints reconstruction: Average absolute error (in degrees) for the manifold structured estimator (SP), the manifold regression (MR) approach in [47] and the KRLS baseline. (Right) Fingerprint reconstruction of a single image where the structured predictor achieves 15.7 of average error while KRLS 25.3.

penalizes especially matrices with very different eigenvalues and our method cannot predict matrices with non-positive eigenvalues, as opposed to KRLS which computes a linear solution in $\mathbb{R}^{d^2}$ and then projects it onto the manifold. However the two methods perform comparably with respect to the squared loss. This is consistent with the fact that our estimator is aware of the natural structure of the output space and uses it profitably during learning.

### 4.3 Fingerprint Reconstruction

We consider the fingerprint reconstruction application in [47] in the context of manifold regression. Given a partial image of a fingerprint, the goal is to reconstruct the contour lines in output. Each fingerprint image is interpreted as a separate structured prediction problem where training input points correspond to the 2D position $x \in \mathbb{R}^2$ of valid contour lines and the output is the local orientation of the contour line, interpreted as a point on the circumference $\mathcal{S}_1$. The space $\mathcal{S}_1$ is a manifold with squared geodesic distance $\triangle_{\mathcal{S}_1}$ between two points $z, y \in \mathcal{S}_1$ corresponding to

$$\triangle_{\mathcal{S}_1}(z, y) = \arccos\left(\langle z, y \rangle\right)^2 \tag{17}$$

where $\arccos$ is the inverse cosine function. In Table 1 we show the computation of the structured loss, the gradient of the structured loss and the chosen retraction for the sphere manifold. We compared the performance of the manifold structured estimator proposed in this paper with the manifold regression approach in [47] on the FVC fingerprint verification challenge dataset[3]. The dataset consists of $48$ fingerprint pictures, each with $\sim 1400$ points for training, $\sim 1000$ points for validation and the rest ($\sim 25000$) for test.

Fig. 1 reports the average absolute error (in degrees) between the true contour orientation and the one estimated by our structured prediction approach (SP), the manifold regression (MR) in [47] and the KRLS baseline. Our method outperforms the MR competitor by a significant margin. As expected, the KRLS baseline is not able to capture the geometry of the output space and has a significantly larger error of the two other approaches. This is also observed on the qualitative plot in Fig. 1 (Left) where the predictions of our SP approach and the KRLS baseline are compared with the ground truth on a single fingerprint. Output orientations are reported for each pixel with a color depending on their orientation (from $0$ to $\pi$). While the KRLS predictions are quite inconsistent, it can be noticed that our estimator is very accurate and even "smoother" than the ground truth.

### 4.4 Multilabel Classification on the Statistical Manifold

We evaluated our algorithm on multilabel prediction problems. In this context the output is an $m$-dimensional histogram, i.e. a discrete probability distribution over $m$ points. We consider as manifold the space of probability distributions over $m$ points, that is the $m$-dimensional simplex $\Delta^m$ endowed with the *Fisher information metric* [3]. We will consider $\mathcal{Y} = \Delta_\epsilon^m$ where we require $y_1, \ldots, y_m \geq \epsilon$, for $\epsilon > 0$. In the experiment we considered $\epsilon = 1e-5$. The geodesic induced by the

Table 3: Area under the curve (AUC) on multilabel benchmark datasets [51] for KRLS and SP.

|          | KRLS | SP (Ours) |
|----------|------|-----------|
| Emotions | 0.63 | **0.73**  |
| CAL500   | **0.92** | **0.92** |
| Scene    | 0.62 | **0.73**  |

Fisher metric is, $d(y, y') = \arccos\left(\sum_{i=1}^{m} \sqrt{y_i y_i'}\right)$ [35]. This geodesic comes from applying the map $\pi \colon \Delta^m \to \mathcal{S}_{m-1}, \ \pi(y) = (\sqrt{y_1}, \dots, \sqrt{y_{m+1}})$ to the points $\{y_i\}_{i=1}^{n} \in \Delta^m$. This results in points that belong to the intersection of the positive quadrant $\mathbb{R}_{++}^m$ and the sphere $\mathcal{S}_{m-1}$. We can therefore use the geodetic distance on the Sphere and gradient and retraction map described in Table 1. We test our approach on some of the benchmark multilabel datasets described in [51] and we compare the results with the KRLS baseline. We cross-validate $\lambda$ and $\sigma$ taking values, respectively, from the intervals $[10^{-6}, 10^{-1}]$ and $[0.1, 10]$. We compute the area under curve (AUC) [45] metric to evaluate the quality of the predictions, results are shown in Table 3.

## 4.5 Additional Example

We conclude this section with a further relevant example of our proposed approach to the setting of applications to relational knowledge, which we plan to investigate in future work. In particular, we consider settings where the output space $\mathcal{M}$ corresponds to the Hyperbolic space (or Poincaré disk), which has recently been adopted by the knowledge representation community to learn embeddings of relational data to encode discrete semantic/hierarchical information [34, 33]. The embedding is such that symbolic objects (e.g. words, entities, concepts) with high semantic or functional similarity are mapped into points with small hyperbolic geodesic distance. Typically, learning the embedding is a time consuming process that requires training from scratch on the whole dataset whenever a new example is provided. To address this issue, with our manifold regression approach we could learn $f : X \to \mathcal{M}$ with $x \in X$ the observed entity and $f(x)$ its predicted embedding. This would allow to transfer the embedding learned using techniques such as those in [34] to new points, without retraining the entire system from scratch. Interestingly, our theory is applicable to this setting since $\mathcal{M}$ satisfies Asm. 1.

## 5 Conclusions

In this paper we studied a structured prediction approach for manifold valued learning problems. In particular we characterized a wide class of loss functions (including the geodesic distance) for which we proved the considered algorithm to be statistically consistent, additionally providing finite sample bounds under standard regularity assumptions. Our experiments show promising results on synthetic and real data using two common manifolds: the positive definite matrices cone and the sphere. With the latter we considered applications on fingerprint reconstruction and multi-labeling. The proposed method leads to some open questions. From a statistical point of view it is of interest how invariants of the manifold explicitly affect the learning rates, see Remark 1. From a more computational perspective, even if experimentally our algorithm achieves good results we did not investigate convergence guarantees in terms of optimization.

**Acknowledgments.**
A. R. acknowledges the support of the European Research Council (grant SEQUOIA 724063). L. R. acknowledges the support of the AFOSR projects FA9550-17-1-0390 and BAA-AFRL-AFOSR-2016-0007 (European Office of Aerospace Research and Development), and the EU H2020-MSCA-RISE project NoMADS - DLV-777826. The work of L. R. and G. M. M. is supported by the Center for Brains, Minds and Machines (CBMM), funded by NSF STC award CCF-1231216, and the Italian Institute of Technology. We gratefully acknowledge the support of NVIDIA Corporation for the donation of the Titan Xp GPUs and the Tesla k40 GPU used for this research. This work was supported in part by EPSRC grant EP/P009069/1.

## Footnotes

[2]This is standard assumption for universal consistency (see [48]). An example of continuous universal kernel on $\mathcal{X} = \mathbb{R}^d$ is the Gaussian $k(x, x') = \exp(-\|x - x'\|^2/\sigma)$, for $\sigma > 0$.

[3] http://bias.csr.unibo.it/fvc2004

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
