[Supplementary Material]

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

# Appendix

The appendix of this work is organized in the following sections:

- A SELF property for smooth functions defined on manifolds (Thm. 1).
- B Proof of SELF property for squared geodesic distances (Thm. 2).
- C Generalization bounds for the structured estimator with squared geodesic loss (Thm. 4).
- D Basic definitions and concepts for Riemannian manifolds.
- E Riemannian gradient descent algorithm.
- F A note on KRLS for the experiments in Section 4.2 on PD matrices.

# A  Proof of Thm. 1

We prove here intermediate results that will be key to prove Thm. 1. We refer to [28] for basic definitions on manifolds and to [4] for an introduction on reproducing kernel Hilbert spaces (RKHS).

**Notation and Definitions.** We recall here basic notations and definition that will be used in the following. Given a smooth manifold $\mathcal{M}$, for any open subset $U \subseteq \mathcal{M}$ we denote by $C^\infty(U)$ the set of smooth functions on $U$ and with $C_0^\infty(U)$ the set of *compactly supported* smooth functions on $U$, namely functions such that the closure of their support is a compact set. For a compact subset $N \subset \mathcal{M}$ we denote by $C_0^\infty(N)$ the set of all functions $h : N \to \mathbb{R}$ that admit an extension $\bar{h} \in C_0^\infty(\mathcal{M})$ such that $\bar{h}|_N = h$ and its support is contained in $N$, namely it vanishes on the border of $N$. Finally, for any subset $N$ of $\mathcal{M}$ we denote $C^\infty(N)$ the set of all functions that admit a smooth extension in $C^\infty(\mathcal{M})$.

In the following, a central role will be played by tensor product of topological vector spaces [54]. In particular, for a Hilbert space $\mathcal{H}$, we will denote $\mathcal{H} \otimes \mathcal{H}$ the closure of the tensor product between $\mathcal{H}$ and itself with respect to the canonical norm such that $\|h \otimes h'\|_{\mathcal{H} \otimes \mathcal{H}} = \|h\|_{\mathcal{H}}\|h'\|_{\mathcal{H}}$ for any $h, h' \in \mathcal{H}$. Moreover, to given a compact set $N \subset \mathbb{R}^d$, we recall that $C_0^\infty(N)\hat{\otimes}_\pi C_0^\infty(N)$ denotes the completion of the topological tensor product between $C_0^\infty(N)$ and itself with respect to the projective topology (see [54] Def. 43.2 and 43.5). In the following, for simplicity, we will denote this space with $C_0^\infty(N) \otimes C_0^\infty(N)$ with some abuse of notation. Finally, for any subset $\mathcal{Y} \subseteq \mathcal{M}$ and space $\mathcal{F}$ of functions from $\mathcal{M}$ to $\mathbb{R}$ we denote by $\mathcal{F}|_{\mathcal{Y}}$ the space of functions from $\mathcal{Y}$ to $\mathbb{R}$ that admit an extension in $\mathcal{F}$. In particular not that $C^\infty(\mathcal{Y}) = C^\infty(\mathcal{M})|_{\mathcal{Y}}$.

## A.1  Auxiliary Results

We are ready to prove the auxiliary results.

**Lemma 5.** *Let $\mathcal{M}$ be a topological space, $Y \subseteq \mathcal{M}$ be a compact subset and $\mathcal{H}$ a reproducing kernel Hilbert space of functions on $\mathcal{M}$ with kernel $K : \mathcal{M} \times \mathcal{M} \to \mathbb{R}$ such that there exists $\kappa > 0$ for which $k(y, y) \leq \kappa^2$ for any $y \in \mathcal{Y}$. Then, for any $\bar{h} \in \mathcal{H} \otimes \mathcal{H}$, its restriction to $\mathcal{Y} \times \mathcal{Y}$, $h = \bar{h}|_{\mathcal{Y} \times \mathcal{Y}}$ is SELF.*

*Proof.* Denote $K_y = k(y, \cdot) \in \mathcal{H}$ for every $y \in \mathcal{M}$. Then the space $\mathcal{H} \otimes \mathcal{H}$ is an RKHS with reproducing kernel $\bar{K} : (\mathcal{M} \times \mathcal{M}) \times (\mathcal{M} \times \mathcal{M}) \to \mathbb{R}$ such that $\bar{K}((y, z), (y', z')) = K(y, y')K(z, z')$ for any $y, y', z, z' \in \mathcal{M}$ (see e.g. [4]). In particular $\bar{K}_{(y,z)} = K_y \otimes K_z$. Let now $\bar{h} : \mathcal{M} \times \mathcal{M} \to \mathbb{R}$ be a function in $\mathcal{H} \otimes \mathcal{H}$. In particular, there exist a $V \in \mathcal{H} \otimes \mathcal{H}$ such that $\langle V, K_y \otimes K_z \rangle_{\mathcal{H} \otimes \mathcal{H}} = \bar{h}(y, z)$ for any $y, z \in \mathcal{Y}$ (reproducing property). Note that $\mathcal{H} \otimes \mathcal{H}$ is isometric to the space of Hilbert-Schmidt operators from $\mathcal{H}$ to itself, with inner product corresponding to $\langle A, B \rangle_{\mathcal{H} \otimes \mathcal{H}} = \langle A, B \rangle_{\mathrm{HS}} = \mathrm{Tr}(A^*B)$ for any $A, B \in \mathcal{H} \otimes \mathcal{H}$, with $A^*$ denoting the conjugate of $A^* \in \mathcal{H} \otimes \mathcal{H}$. Therefore, for any $y, z \in \mathcal{Y}$ we have

$$\bar{h}|_{\mathcal{Y} \times \mathcal{Y}}(y, z) = \bar{h}(y, z) = \langle V, K_y \otimes K_z \rangle_{\mathcal{H} \otimes \mathcal{H}} = \mathrm{Tr}(V^* K_y \otimes K_z) = \langle K_z, V^* K_y \rangle_{\mathcal{H}}. \tag{18}$$

Since $K_y$ is bounded in $\mathcal{H}$, for $y \in \mathcal{Y}$ and the operator norm of $V$ is bounded by its Hilbert-Schmidt norm, namely $\|V\| \leq \|V\|_{\mathrm{HS}}$, we can conclude that $h = \bar{h}|_{\mathcal{Y} \times \mathcal{Y}}$ is indeed SELF. $\square$

**Lemma 6.** *Let $\mathcal{M}$ satisfy Asm. 1. Then there exists a reproducing kernel Hilbert space of functions $\mathcal{H}$ on $\mathcal{M}$, with bounded kernel, such that $C_0^\infty(\mathcal{M}) \subseteq \mathcal{H}$.*

*Proof.* Let $H_s^2(\mathcal{M})$ denote the Sobolev space on $\mathcal{M}$ of squared integrable functions with smoothness $s > 0$ (see [23] for the definition of Sobolev spaces on Riemannian manifolds). By construction (see page 47 of [23]), $C_0^\infty(\mathcal{M}) \subset H_s^2(\mathcal{M})$ for any $s > 0$. To prove this Lemma, we will show that $H_s^2(\mathcal{M})$ is an RKHS for any $s > d/2$. The proof is organized in two steps.

**Step 1: $H_s^2(\mathcal{M})$ is continuously embedded in $C(\mathcal{M})$.** By Asm. 1, we can apply Thm. 3.4 in [23] (see also Thm. 2.7 [23] for compact manifolds), which guarantees the existence of a constant $C > 0$ (see last lines of the proofs for its explicit definition) such that

$$\sup_{y \in \mathcal{M}} |f(y)| \leq C \|f\|_{\mathcal{H}_s^2(\mathcal{M})},$$

for any $y \in \mathcal{M}$ and $f \in \mathcal{H}_s^2(\mathcal{M})$.

**Step 2: Constructing $\mathcal{H}$ from $H_s^2(\mathcal{M})$.** Prop. 2.1 of [23] proves that there exists an inner product, that we denote by $\langle \cdot, \cdot \rangle_{\mathcal{H}}$, whose associated norm is equivalent to $\| \cdot \|_{H_s^2(\mathcal{M})}$ and such that the space $\mathcal{H} = (H_s^2(\mathcal{M}), \langle \cdot, \cdot \rangle_{\mathcal{H}})$ is a Hilbert space.

Now, for any $y \in \mathcal{M}$ denote by $e_y : \mathcal{H} \to \mathbb{R}$, the linear functional corresponding to the evaluation, that is $e_y(f) = f(y)$. Now by Step 1, we have that the linear functional $e_y$ is uniformly bounded and so continuous, indeed,

$$|e_y(f)| = |f(y)| \leq C \|f\|_{\mathcal{H}}, \quad \forall f \in \mathcal{H}.$$

So by the Riesz representation theorem $e_y \in \mathcal{H}$ and so $\mathcal{H}$ is a reproducing kernel Hilbert space, with kernel $k(y, y') = \langle e_y, e_{y'} \rangle_{\mathcal{H}}$, (see [4], page 343, for more details). Note finally that the kernel is bounded since

$$\|e_y\|_{\mathcal{H}} = \sup_{\|f\|_{\mathcal{H}} \leq 1} |\langle e_y, f \rangle_{\mathcal{H}}| = \sup_{\|f\|_{\mathcal{H}} \leq 1} |e_y(f)| \leq C,$$

and therefore $k(y, y') \leq \|e_y\|_{\mathcal{H}} \|e_{y'}\|_{\mathcal{H}} \leq C^2$. $\qquad\square$

In the following, let $A \subseteq \{f : U \to S\}$ and $B \subseteq \{g : V \to S\}$, with $U, V, S$ topological spaces. We denote $A \cong B$ if there exists an invertible map $q : U \to V$, such that $B = A \circ q^{-1}$ and $A = B \circ q$.

**Lemma 7** (see also [34, 35])**.** *Let $U$ be a geodesically convex open subset of a $d$-dimensional complete Riemannian manifold $\mathcal{M}$ without border, then there exists a smooth map $q : U \to \mathbb{R}^d$ with smooth inverse, such that*

$$C^\infty(U) \cong C^\infty(\mathbb{R}^d), \qquad and \qquad C_0^\infty(U) \cong C_0^\infty(\mathbb{R}^d)$$

*moreover for any compact set $\mathcal{Y} \subset U$ there exists a compact set $R \subset \mathbb{R}^d$ such that $R = q(\mathcal{Y})$ and the map $s$, that is the restriction of $q$ to $\mathcal{Y} \to R$, guarantees*

$$C^\infty(\mathcal{Y}) \cong C^\infty(R), \qquad and \qquad C_0^\infty(\mathcal{Y}) \cong C_0^\infty(R)$$

*Proof.* By Lemma 9, there exists a point $p \in U$ such that $d(p, \cdot)$ admits all directional derivatives in all points $q \in U$ (it is, in fact in $C^\infty(U)$). We are therefore in the hypotheses of Thm. 2 in [57], from which we conclude that there exists a smooth diffeomorphism between $U$ and $\mathbb{R}^d$ (with smooth inverse). Denoting by $q$ the diffeomorphism between $U$ and $\mathbb{R}^d$, for any function $f \in C^\infty(U)$, we have $f \circ q^{-1} \in C^\infty(\mathbb{R}^d)$, so $C^\infty(U) \circ q^{-1} \subseteq C^\infty(\mathbb{R}^d)$ and for any function $g \in C^\infty(\mathbb{R}^d)$ we have $g \circ q \in C^\infty(U)$, so $C^\infty(\mathbb{R}^d) \circ q \subseteq C^\infty(U)$. Finally we recall that if $A \subseteq B$, then $A \circ p \subseteq B \circ p$ for any set $A, B$ and any map $p$ applicable to $A, B$. Then

$$C^\infty(U) = C^\infty(U) \circ q^{-1} \circ q \subseteq C^\infty(\mathbb{R}^d) \circ q \subseteq C^\infty(U)$$

and so $C^\infty(N) \cong C^\infty(\mathbb{R}^d)$. The same reasoning holds $C_0^\infty(U) \cong C_0^\infty(\mathbb{R}^d)$.

Analogously, the smooth diffeomorphism $q$ maps compact subsets of $U$ to compact subsets of $\mathbb{R}^d$. Denote by $R \subset \mathbb{R}^d$ the compact subset that is $q(\mathcal{Y})$, the image of $\mathcal{Y} \subseteq U$ a compact subset of $U$, then $s$ is the restriction of $q$ to $\mathcal{Y} \to R$. By the same reasoning as above, we have that $C^\infty(\mathcal{Y}) \cong C^\infty(R)$ via $s$. $\qquad\square$

**Lemma 8.** *Let $U$ be a open geodesically convex subset of a complete Riemannian $d$-dimensional manifold $\mathcal{M}$ and $\mathcal{Y}$ a compact subset of $U$, then there exists a compact subset $N \subseteq U$ such that $\mathcal{Y}$ belongs to the interior of $N$ and*

$$C^\infty(\mathcal{Y} \times \mathcal{Y}) \subseteq (C_0^\infty(N) \otimes C_0^\infty(N))|_{Y \times Y}.$$

*Moreover, $C^\infty(\mathcal{Y}) \subseteq C_0^\infty(N)|_{\mathcal{Y}}$.*

*Proof.* We first consider the real case $U = \mathcal{M} = \mathbb{R}^d$ with Euclidean metric. By Cor. 2.19 in [28], for any open subset $V \subset \mathbb{R}^d$ we have that any $f \in C^\infty(\mathcal{Y})$ admits an extension $\tilde{f} \in C^\infty(\mathbb{R}^d)$ such that $\tilde{f}|_{\mathcal{Y}} = f$ and $\operatorname{supp} \tilde{f} \subset C_0^\infty(V)$. Then, since $\mathcal{Y}$ is bounded (compact in a complete space), there exists a bounded open set $V$ containing $\mathcal{Y}$. Let $N = \overline{V}$ the closure of $V$. $N$ is a compact set as well and contains $\mathcal{Y}$ in its interior. In particular, since for any $f \in C^\infty(\mathcal{Y})$ the extension $\tilde{f}$ has support contained in $V \subset N$, this shows that $C^\infty(\mathcal{Y}) \subseteq C_0^\infty(N)$. Analogously we have $C^\infty(\mathcal{Y} \times \mathcal{Y}) \subseteq C_0^\infty(N \times N)$.

Now, by Thm. $51.6$ $(a)$ in [54], we have that

$$C_0^\infty(N) \otimes C_0^\infty(N) \cong C_0^\infty(N \times N). \tag{19}$$

which concludes the proof in the real setting. The proof generalizes trivially to the case where $U$ is an open geodesically convex subset of a complete Riemannian manifold thanks to the isomorphisms between spaces of smooth functions provided by Lemma 7. □

### A.2  Proof of Thm. 1

For the following results we need to introduce the concept of *cut locus*. For any $y \in \mathcal{M}$, denote by $\operatorname{Cut}(y) \subseteq \mathcal{M}$ the *cut locus* of $y$ the closure of the set of points $z \in \mathcal{M}$ that are connected to $y$ by more than one minimal geodesic (see [20, 46]). For any $y \in \mathcal{Y}$ we have $y \in \mathcal{M} \setminus \operatorname{Cut}(y)$, see e.g. Lemma $4.4$ in [46].

Finally we refine Asm. 2 to avoid pathological cases. Indeed a geodesically convex set can still have conjugate points on the boundary. To avoid this situation we restate Asm. 2 as follows

**Assumption 2'** $\widetilde{M}$ *is an open geodesically convex subset of the manifold $\mathcal{M}$ and $\mathcal{Y}$ is a compact subset of $\widetilde{M}$.*

*Proof of Thm. 1.* By Asm. 2', let $\widetilde{\mathcal{M}}$ be an open geodesically convex subset of $\mathcal{M}$ such that $\mathcal{Y} \subset \widetilde{\mathcal{M}} \subseteq \mathcal{M}$. Apply Lemma 8 and let $N \subseteq \widetilde{\mathcal{M}}$ be a compact set such that $\mathcal{Y}$ is contained in the interior of $N$, namely

$$C^\infty(\mathcal{Y}) \subseteq C_0^\infty(N)|_{\mathcal{Y}} \subseteq C_0^\infty(\mathcal{M})|_{\mathcal{Y}} \subseteq \mathcal{H}|_{\mathcal{Y}}. \tag{20}$$

Then, by applying again Lemma 8 we have

$$C^\infty(\mathcal{Y} \times \mathcal{Y}) \subseteq (C_0^\infty(N) \otimes C_0^\infty(N))|_{\mathcal{Y} \times \mathcal{Y}} \subseteq (\mathcal{H} \otimes \mathcal{H})|_{\mathcal{Y} \times \mathcal{Y}}. \tag{21}$$

Therefore we conclude that for any $h \in C^\infty(\mathcal{Y})$, there exists $\bar{h} : \mathcal{M} \times \mathcal{M} \to \mathbb{R}$ with $\bar{h} \in \mathcal{H} \otimes \mathcal{H}$ and $h = \bar{h}|_{\mathcal{Y} \times \mathcal{Y}}$. Finally we apply Lemma 5 to $\bar{h}$, which guarantees $h$ to be SELF. □

## B  Proof of Thm. 2

We prove a preliminary result.

**Lemma 9.** *Let $\mathcal{M}$ be a Riemannian manifold and $N$ be a geodesically convex subset of $\mathcal{M}$. Then,*

$$d^2|_{N \times N} \in C^\infty(N \times N).$$

*Proof.* For any $y \in \mathcal{M}$, denote $\operatorname{Cut}(y) \subseteq \mathcal{M}$ the *cut locus* of $y$, that is the set of points in $z \in \mathcal{M}$ that are connected by more than one minimal geodesic curve with $y$ (see [20, 46]). Let $\operatorname{Cut}(\mathcal{M}) =$

$\bigcup_{y \in \mathcal{M}} (\{y\} \times \mathrm{Cut}(y)) \subseteq \mathcal{M} \times \mathcal{M}$. Then, then the squared geodesic distance is such that (see e.g. [56], page 336)

$$d^2 \in C^\infty(\mathcal{M} \times \mathcal{M} \setminus \mathrm{Cut}(\mathcal{M})).$$

Now note that by definition of geodesically convex subset $N \subseteq \mathcal{M}$, for any two points in $N$ there exist one and only one minimizing geodesic curve connecting them. Therefore, $N \times N \cap \mathrm{Cut}(\mathcal{M}) = \emptyset$ and consequently $N \times N \subseteq \mathcal{M} \times \mathcal{M} \setminus \mathrm{Cut}(\mathcal{M})$. We conclude that the restriction of $d^2$ on $N \times N$ is $C^\infty$ as required. $\qquad\square$

*Proof of Thm. 2.* By Lemma 9, under Asm. 1 and Asm. 2, the squared geodesic distances is smooth. The desired result is then obtained by applying Thm. 1. $\qquad\square$

## C  Proof of Thm. 4

*Proof.* The theorem is proved by combining Thm. 1 with Thm. 5 in [13]. To characterize the constant $\mathsf{c}_\triangle$ we need an extra step.

Under Asm. 1 and Asm. 2 and the smoothness of $\triangle$, we can apply Thm. 1, which characterizes $\triangle$ as SELF. According to the proof of Thm. 1 and in particular of Lemma 5, for any $y, z \in \mathcal{Y}$ we have

$$\triangle(y, z) = \langle \psi(y), V\psi(z) \rangle_\mathcal{H} \tag{22}$$

where $\mathcal{H} = H_s^2(\mathcal{M})$ with $s > d/2$, $\psi(y) = K_y(\cdot)$ where $K : \mathcal{M} \times \mathcal{M} \to \mathbb{R}$ is the reproducing kernel associated to $\mathcal{H}$ and $V : \mathcal{H} \to \mathcal{H}$ is the operator defined in Eq. (18). In particular, by the isometry between the tensor space $\mathcal{H} \otimes \mathcal{H}$ and the space of Hilbert-Schmidt operators from $\mathcal{H}$ to $\mathcal{H}$, we have

$$\|V\|_{\mathrm{HS}} = \|\triangle\|_{\mathcal{H} \otimes \mathcal{H}}. \tag{23}$$

To conclude, since $\triangle$ is SELF, the following generalization bound in Thm. 5 from [13]

$$\mathcal{E}(\widehat{f}) - \mathcal{E}(f^*) \leq \|V\| \, \mathsf{q} \, \tau^2 \, n^{-\frac{1}{4}} \tag{24}$$

holds with probability at least $1 - 8e^{-\tau}$. Here, $\|V\|$ denotes the operator norm of $\|V\|$ and $\mathsf{q}$ is a constant depending only on $\mathcal{Y}$ and the distribution $\rho$ (see end of proof of Lemma 18 for additional details). Finally, we recall, by the relation between the operator and Hilbert-Schmidt norm, that $\|V\| \leq \|V\|_{\mathrm{HS}} = \|\triangle\|_{\mathcal{H} \otimes \mathcal{H}} = \mathsf{c}_\triangle$.

$\qquad\square$

## D  Additional Comparison

We have implemented the approach in [50, 51] and tested it on the synthetic data in Section 4.2. We note that these works were the only ones we found in the literature that address the problem of manifold-valued regression. We also note in passing that the seminal work of Steinke, Hein and Schölkopf [50, 51] did focus on proposing and solving an ERM strategy for manifold-valued learning, but did not study the theoretical properties of the resulting estimator.

The method essentially consists of a Nadaraya-Watson-like estimator combined with an iterative projection over the output manifold. In our implementation of [51] we used Gaussian kernel and first order polynomials as interpolating functions (see Sec. 4 in [51] for notation and details). Note that in general it is not always possible to uniformly sample the centers for the interpolating functions over the output manifold (as suggested in [51]). Thus, we used the training outputs as centers. To train the model, the parameters of the first order polynomials have been optimized using a quasi-Newton method on the empirical risk with geodesic distance and *Eells energy* regularizer.

| $\triangle_{PD}$ err. | Krls | Ours | [29] | | time (s) | train | test |
|---|---|---|---|---|---|---|---|
| $n = 10$ | 44 | **1.24** | 6.98 | | KRLS | 0.2 | 0.01 |
| $n = 20$ | 59 | **1.33** | 11.2 | | OURS | 0.2 | 9.8 |
| $n = 30$ | 67 | **1.55** | 18.1 | | [29] | 1740 | 0.1 |

Figure 2: Pictorial representations of the exponential map.

The table above (Left) extends Table 2 adding the accuracy of [29] on the synthetic experiment. The table on the right reports the computational times of both our approach and [51] for $n = 30$ and PD matrices of dimension $d = 30$ (hence manifold dimension $O(d^2)$). The long training times for [51] are due to the high-dimensional non-linear optimization required to learn the weights parametrizing the interpolating function (which are of order $O(d^2 n)$, here $\sim 30000$). Note that the complexity of inference scales proportionally (indeed linearly) with the number $n$ of training points and corresponds to finding the barycenter of $n$ points on the output manifold. Stochastic methods such as Riemannian SGD (R-SGD) can be adopted to obtain approximate predictions that are still within the statistical learning bound of Thm. 4, but require less computations. We will: 1) add a remark at the end of Sec 4.1 detailing the impact of adopting a stochastic algorithm in this setting; 2) Add a plot comparing time vs accuracy of Riemannian GD vs R-SGD on the synthetic experiments for different values of $n$ and manifold dimension $d$. 3) Add the performance of R-SGD for the real experiments in Sec. 4.3 (Table 3 and Fig. 1).

We observe that our approach is significantly more accurate than [51]. Admittedly, it may be possible that better performance could be achieved with [51] by performing a finer cross-validation. However, in practice, the long computational times required to train this method make it prohibitive to perform extensive model-selection. For these reasons we did not run [51] on the multi-labeling tasks.

## E   Differential geometry definitions

A Riemannian manifold $(\mathcal{M}, g)$ of dimension $n$ is a topological space $\mathcal{M}$ such that every point $y \in \mathcal{M}$ has a neighbourhood which is homeomorphic to an open set in Euclidean space $\mathbb{R}^n$ and $g$ is a collection of inner product defined in every tangent space $T_y\mathcal{M}$ of every point $y \in \mathcal{M}$. Intuitively, the tangent space $T_y\mathcal{M}$ is an approximation of a neighbourhood of $y \in M$ that has a vector space structure. We will denote the inner product of $u, v \in T_y\mathcal{M}$ as $\langle u, v \rangle_y$. Thanks the inner product structure in every tangent space of the manifold we can compute gradients of functions $f \colon \mathcal{M} \to \mathbb{R}$ that we will denote with $\nabla_{\mathcal{M}} f \colon \mathcal{F}_y(\mathcal{M}) \to T_y\mathcal{M}$. Where $\mathcal{F}_y(\mathcal{M})$ is the set of smooth real-valued functions defined on a neighbourhood of $y$.

For any $y_0, y_1 \in \mathcal{M}$ and $v \in T_y\mathcal{M}$ there is a unique smooth *geodesic* curve $\gamma \colon [0, 1] \to \mathcal{M}$ such that $\gamma(0) = y_0$, $\gamma(1) = y_1$ and $\frac{d}{dt}\gamma(0) = v$, this curve locally minimizes the path between $y_0$ and $y_1$. Given the geodesic between $y_0$ and $y_1$ with derivative $\frac{d}{dt}\gamma(0) = v$, the exponential map $Exp_{y_0} \colon T_{y_0}\mathcal{M} \to \mathcal{M}$ maps vector $v \in T_{y_0}\mathcal{M}$ to $y_1$. A retraction $R_y \colon T_y\mathcal{M} \to \mathcal{M}$, is a first order approximation of the exponential map. Exponential maps are retractions.

## F   Riemannian Gradient Descent

In this section we report fully the algorithm Riemannian Gradient Descent.

---

**Algorithm 1** Riemannian gradient descent

---

**Require:** number of iterations $T$, step size $\eta$, initial point $y_0$
1: **for** $t = 0, \ldots, T-1$ **do**
2:      $v_t = \nabla_{\mathcal{M}} \sum_{i=1}^{n} \alpha_i(x) \triangle (y_t, y_i)$
3:      $y_{t+1} \leftarrow R_{y_t}(\eta_t \, v_t)$
4: **end for**
5: **return** $y_T$

---

## G    Kernel Regularized Least Squares estimator for Positive definite matrices

We consider the case where we want to use KRLS estimators to predict a positive definite matrix given a data set $\{x_i, y_i\}_{i=1}^{n}$. The KRLS estimator $f: \mathbb{R}^d \to \mathbb{R}$ is a function defined as $f(x) = \sum_{i=1}^{n} k(x, x_i) w_i$, where $k: \mathbb{R}^d \times \mathbb{R}^d \to \mathbb{R}$ is a reproducing kernel and $w = [w_1, \ldots, w_n] \in \mathbb{R}^n$ are constant weights computed by solving the problem:

$$\min_{f \in \mathcal{H}} \frac{1}{n} \sum_{i=1}^{N} \|\hat{y}_i - \hat{K}w\|^2 + \lambda \|w\|^2$$

$\hat{K} \in \mathbb{R}^{n \times n}$ is the kernel matrix whose elements are defined as $(K)_{ij} = k(x_i, x_j)$.

To predict a positive definite matrix $y \in \mathbb{P}_d^{++}$, a KRLS estimator is learned for every element of the flattened matrix $vec(y) \in \mathbb{R}^{d^2}$. Suppose $j \in \{1, \ldots, d^2\}$ is the index of the $j$-th component of $vec(y)$ that we want to predict, then we want to learn the estimator $f^{(j)}(x) = \sum_{i=1}^{n} k(x, x_i) w_i^{(j)}$. The corresponding problem has labels $\hat{y}^{(j)} = [vec(y_1)_j, \ldots, vec(y_{d^2})_j]$ and we solve for $w^{(j)} = [w_1^{(j)}, \ldots, w_n^{(j)}]$. Indeed we compute $d^2$ estimator to predict $vec(f) = [f^{(1)}(x), \ldots, f^{(d^2)}]$ and then recover $y$ from its vectorized form. Once the matrix is predicted we enforce it to be positive definite by performing a spectral decomposition and setting the negative eigenvalues to a small positive constant.

In general, when doing structured predictions with KRLS approach, it is necessary to project the outcome of the prediction on the desired manifold.