[Reviews · NeurIPS 2018]

Reviewer 1



Summary: This paper is an extension of the results presented in “A Consistent Regularization Approach for Structured Prediction” by Ciliberto et al. It focuses on the specific case where the output space is a Riemannian manifold, and describes/proves sufficient conditions for loss functions defined over manifolds to have the properties of what is called a “Structure Encoding Loss Function” (SELF). Ciliberto et al presents an estimator that, when used with a SELF, has provable universal consistency and learning rates; this paper extends this estimator and these prior theoretical results to be used also with the aforementioned class of loss functions defined over manifolds, with a specific focus placed on the squared geodesic distance. After describing how inference can be achieved using the previously defined estimator for the specific output spaces defined here, experiments are run on a synthetic dataset with the goal of learning the inverse function over the set of positive-definite matrices and a real dataset consisting of fingerprint reconstruction. Comments: This work is well-written and well-organized, and it is easy to follow all of the concepts being presented. Furthermore, its primary achievement - further characterizing a set of output spaces/functions for which previous methods can be applied - provides a useful extension of that work. The primary criticism I have of this paper is that it is somewhat incremental in nature - the SELF framework, corresponding consistency and learning rate results, and estimator are all borrowed from Ciliberto et. al. In contrast, all this paper does is describe one set of output spaces/losses for which those previous results can be applied. This might be more forgivable if it was then shown that these methods, applied to many examples of problems satisfying the required assumptions/properties, prove themselves to be very useful. However, though the experiments provide interesting preliminary results, only one task consisting of real data was used to demonstrate the utility of the described approach. At the very least, more examples need to be provided of tasks whose output spaces satisfy assumptions 1 and 2 that are not discrete or euclidean - but it would be preferred if more complete experimentation was done for more tasks. Miscellaneous: -Line 60: “we consider a structured prediction approach to ma following…” -Line 82: “section section 3” With further experimentation on a variety of tasks on which other methods may struggle, this paper would make an excellent contribution. As it is currently, however, I think it is a borderline case. I am leaning towards the side of accepting it since everything that is presented is useful and interesting. POST AUTHOR RESPONSE UPDATE: Thanks for the response. The comparison against the other baseline definitely improves the quality of the synthetic experiment. Additionally, I am glad that you are able to provide another example that will clarify the utility of the proposed methods. Regardless, the point still stands that your experimental comparisons are somewhat weak. If were able to provide interesting results for this new task, then I might have increased your score, but as the work stands I feel like the score I originally gave is still appropriate.

Reviewer 2



- The authors adapt the framework from [8] (A consistent regularization approach for structured prediction) to solve the problem of manifold regression. The framework from [8] allows the manifold regression solution to be represented as a minimizer of the linear combination of loss over training samples (similar to the kernel trick). Their main theoretical contribution is to show that the Hilbert space condition required by [8] over the loss function is satisfied by the squared geodesic distance. This part is novel and interesting. - The part on generalization bounds and consistency is highly similar to [8], and is a relatively small contribution. - Experiments on learning synthetic positive definite matrices, and two real datasets with fingerprint reconstruction and multilabel classification, show the method performs better than previous approaches. However, there is no comparison with manifold regression (MR[30]) on the matrix experiments and the multilabel classification experiments. It is easy to beat kernel least square because it doesn't take into account the manifold structure, so it would be good to provide comparison with manifold-based methods on these two sets of experiments. - In terms of efficiency, what is the runtime of the method in training and prediction? The prediction function in equation (2) grows in complexity with the training set size, and performing iterative optimization on it as in section 4.1 can be quite expensive.

Reviewer 3



Summary: A manifold valued regression is formulated based on a framework of structured data prediction. The author extends theoretical framework proposed for the discrete structured prediction to a continuous scenario of the manifold valued output. Overall evaluation: The analysis for the manifold prediction is interesting, which seemingly has not been widely studied. The experimental verification is slightly weak. Other comments: - The experimental comparison is mainly with kernel regularized least squares (KRLS), but KRLS is a quite general regression method, and thus it is not clear if KRLS is appropriate baseline for each specific task. - Section 4.2 is not convincing for me. The baseline method should use the same loss function. As the author pointed out, KRLS does not optimize (15) and thus it is trivial that the proposed method show better accuracy in terms of (15). Since the loss function (15) itself would not be author's novelty, I don't think the results in Table 2 is meaningful comparison. A possible comparison would be comparing with some other model optimized by the same loss function (15).